

# High growth potential and activity of 0.2 μm filterable bacteria habitually present in coastal seawater

Yumiko Obayashi[1], Satoru Suzuki[1]

[1]Center for Marine Environmental Studies, Ehime University, Matsuyama 790-8577, Japan

*Correspondence to*: Yumiko Obayashi (obayashi.yumiko.nn@ehime-u.ac.jp)

**Abstract.** The presence of 0.2 μm filterable bacteria in aquatic environments has been known. Some of these bacteria have been reported to be starvation forms, especially those in oligotrophic oceanic seawater. However, 0.2 μm filterable bacteria have not yet been described in temperate coastal seawater. Here, we report the presence of 0.2 μm filterable bacteria in coastal seawater with their high growth potential that appeared under the absence of grazers. In this study, filtered seawater

(FSW) microcosms were prepared with 0.2 μm filtered coastal seawater collected in summer and winter without any nutritional amendment and incubated at the ambient seawater temperature (25°C in summer, 12°C in winter) and lower temperature (6°C). During the first several days of the incubations of FSW, the number of prokaryotes collected on 0.2 μm filters markedly increased especially at the ambient seawater temperatures. The diversity of the regenerated bacteria community was comparable to that of the original bacterial community, and most of the identified bacteria were typical

marine bacteria (members of Alphaproteobacteria, Gammaproteobacteria and Bacteroidetes), indicating that the 0.2 μm filterable forms of typical marine bacteria show rapid growth under the no grazing and low competition conditions present in the FSW bottles. These results suggest that 0.2 μm filterable bacteria are habitually present even in coastal water, and that these bacteria are always ready for growing in changeable aquatic ecosystems.

## 1 Introduction

The importance of heterotrophic prokaryotes in marine aquatic ecosystems has been recognized in terms of microbial food web and biogeochemical carbon cycle (Azam and Malfatti, 2007). In addition to bacterial abundance and community structure in marine ecosystems, bacterial growth and mortality rates, heterotrophic bacterial production, respiration, and activities of extracellular hydrolytic enzymes in seawater have been investigated as efforts to quantify microbiological processes in aquatic environment (e.g., Alonso-Sáez et al., 2008). Bacterial abundance in the environment is generally

determined by the balance of the growth rate, which is thought to be controlled by nutrient availability, and the mortality rate, which is thought to be defined by grazing of bacterivorous heterotrophs (mostly protistan microzooplankton) and cell lysis due to viral infection. It is difficult to separately estimate the in situ bacterial growth rate and mortality rate.



For investigations of abundance and community structure of prokaryotes, water samples are typically filtered on 0.2 – 0.45 µm pore-size filters; however, microbes in 0.2 µm filtrate have been reported from various aquatic environments (e.g., MacDonell and Hood, 1982; Hahn et al., 2003; Miteva and Brenchley, 2005). These filterable bacteria may actually be smaller than 0.2 µm (ultrasmall cells) or are larger but flexible cells that can pass through the pores of the filter. Wang et al.

reported that not only cell size but also their shape and flexibility affect filterability (Wang et al., 2007; 2008). Based on cultivation of filterable bacteria from aquatic environments, two types of ultrasmall cells have been identified (Torrella and Morita, 1981): ones that have larger cell size (increase in cell size) when incubated with adequate nutrients despite that they were originally from the 0.2 µm filtrates (Torrella and Morita, 1981; Vybrial et al., 1991), and the other ones that did not increase in size (obligate ultramicrobacteria) even after incubation with nutrient-rich media (Torrella and Morita, 1981; Hahn

et al., 2003). The former has been considered to be the starvation forms of these bacteria in natural oligotrophic aquatic habitats (seawater, freshwater). Phylogenetic analyses of 0.2 µm filterable prokaryotes in various oligotrophic natural aquatic environments have also been conducted using a culture-independent method using 0.1 µm filters to trap the 0.2 µm filterable microbes (Haller et al., 1999; Miyoshi et al., 2005; Naganuma et al., 2007; Nakai et al., 2011; 2013). Haller et al. (1999) reported that most of the 0.2 µm filterable bacteria from the western Mediterranean Sea were starvation forms of typical

marine bacteria, based on a comparison of the profiles of denaturing gradient gel electrophoresis (DGGE) and sequencing of 16S rRNA gene fragments from the >0.2 µm community and the 0.1–0.2 µm community.

From these previous investigations, the presence of 0.2 µm filterable small starvation forms of bacteria in very oligotrophic water such as the open ocean has been recognized. However, 0.2 µm filterable bacteria in temperate coastal seawater have not yet been described. It can be hypothesized that 0.2 µm filterable small starvation forms of bacteria are members of

20 microbial loop also in coastal aquatic ecosystems and that they have high potential for organic matter processing.

In this study, we conducted microcosm experiments using 0.2 µm filtered and unfiltered coastal seawater and monitored changes in bacterial abundance, diversity, community structure, and extracellular hydrolytic enzyme activities in the microcosms. We originally designed these microcosms to estimate the lifetime of "dissolved" extracellular hydrolytic enzyme activity in seawater; however, we found very high growth and high activity of 0.2 µm filterable bacteria in the

25 filtered coastal seawater. The present paper describes the potential activities of the microbial community that were usually disguised due to coexistence with grazers.

## 2 Materials and Methods

### 2.1 Microcosm experiments with unfiltered seawater (UNF) and filtered seawater (FSW)

Two microcosm experiments, Experiment I (Exp I) and Experiment II (Exp II), were conducted in summer (September

2008) and in winter (February 2009), respectively, with surface seawater collected at coast of Matsuyama, Ehime Prefecture,



Japan. Seawater for the experiments was collected by a plastic bucket and transferred to polycarbonate bottles through 150 µm-nylon mesh.

A subset of collected seawater was gently filtered (<0.01 MPa) through 0.2 µm pore-size polycarbonate Nuclepore filters (Whatman) and used as 0.2 µm filtered seawater. Microcosms were prepared with unfiltered seawater (UNF) and 0.2 µm

filtered seawater (FSW), respectively, in 1L polycarbonate bottles. Microcosm incubation was conducted at original seawater temperature (Exp I, 25°C; Exp II, 12°C) and at lower temperature (6°C) in the dark. As a control, autoclaved seawater was also incubated at each temperature. For Exp II, twice filtered seawater (wFSW) was prepared by passing FSW through a second 0.2 µm Nuclepore filter and was also incubated.

Samples for analysis of time-course changes in extracellular enzyme activity, prokaryotic cell number and community

structure were withdrawn from each incubation bottle on Days 1, 2, 3, 6 and 10 for Exp I, and on Days 0, 1, 3, 5, 8, 13 and 19 for Exp II. For Exp I, the Day 0 sample was taken from prepared UNF water and FSW (not from individual incubation bottles). For Exp I, samples for prokaryotic community structure on Day 115 was also taken.

### 2.2 Prokaryotic cell counts

For prokaryotic cell counts, samples from Days 1, 3, 10 for Exp I and Days 0, 3, 8, 19 for Exp II were fixed with neutralized

formaldehyde (final concentration 2%) and stained with 4,6-diamidino-2-phenylindole (DAPI). Two mL of fixed samples were filtered onto 0.2-µm black Nuclepore filters, and counts were made under epifluorescence microscopy (×1000, Olympus BX51).

### 2.3 Extracellular enzyme activities

Potential proteolytic enzyme activity in each freshly withdrawn sample was assayed using the following peptide analog 4-

20 methyl-coumaryl-7-amide (MCA) substrates (Peptide Institute): 3 for aminopeptidase (Arg-MCA, Leu-MCA, Ala-MCA), 2 for trypsin (Boc-Phe-Ser-Arg-MCA, Boc-Leu-Ser-Thr-Arg-MCA), and 2 for chymotrypsin (Suc-Ala-Ala-Pro-Phe-MCA, Suc-Leu-Leu-Val-Tyr-MCA). Potential enzymatic activity at 25°C were measured as described previously (Obayashi and Suzuki, 2005; 2008; Obayashi et al., 2010) with the following modifications: the sample and the MCA substrate were mixed in a black low-binding microplate (Nunc) and fluorescence was measured using a microplate reader (Corona SH8100Lab)

with excitation/emission wavelengths of 380/440 nm for several times with interval ($t_0$, $t_1$, $t_2$, $t_3$) to detect the change of fluorescence intensity. The assay was performed in triplicate for each sample and substrate.

### 2.4 DNA extraction

To analyze the bacterial community structure, DNA was extracted for analysis by polymerase chain reaction (PCR) denaturing gradient gel electrophoresis (DGGE) targeting 16S rRNA gene. In Exp I, samples were not taken from FSW

bottles on Days 1, 2, and 3, and were taken on Days 6, 10 and 115. For each sampling, 70 mL of microcosm sample was





filtered onto 0.2 μm Nuclepore filters and kept at -80°C until DNA extraction by the method of Takasu et al. (2011) without the freeze-thaw steps and with other slight modification. Briefly, the filter was cut into small pieces, soaked in 700 μL of extraction buffer (mixture of 652 μL TE buffer (10 mM Tris, 1 mM EDTA, pH 8.0), 40 μL 10% SDS, 4 μL proteinase K (20 mg mL$^{-1}$) and 4 μL RNase (10 mg mL$^{-1}$)) in a 2 mL tube and incubated at 37°C for 1 h, followed by the addition of 100 μL of 10% CTAB/0.7 M NaCl solution and incubated at 65°C for 10 min. Then equal volume (800 μL) of PCI (phenol: chloroform: isoamyl alcohol = 25:24:1 (v:v:v)) was added into the tube. After mixing and centrifugation (21600 ×g, 4°C, 5 min), the upper aqueous layer was carefully transferred to a fresh 2 mL tube. Equal volume (approximately 800 μL) of CI (chloroform: isoamyl alcohol = 24:1 (v:v)) was added to the tube, and the tube was inverted to mix. After centrifugation (21600 ×g, 4°C, 5 min), the upper layer was transferred to a fresh 1.5 mL tube, and 0.1 volumes of 3 M sodium acetate and 0.6 volumes of isopropanol were added. The sample tubes were then mixed by inversion and stored at 4°C for 1 h until the DNA precipitated. The precipitated DNA was then recovered by centrifugation (21600 ×g, 4°C, 20 min). Pellets were washed with cold 70% ethanol, vacuum dried, and dissolved in 30 μL of sterile ultra-pure water.

**2.5 PCR-DGGE**

Bacterial 16S rRNA gene fragments were amplified by PCR in 100 μL reaction mixtures containing 0.25 μM each of forward primer, 341f-GC-clamp (5'-CGC CCG CCG CGC CCC GCG CCC GCG CCC GTC CGC CGC CCC CCG CCC TAC GGG AGG CAG CAG-3'), and reverse primer, 907r (5'-CCG TCA ATT CMT TTG AGT TT-3'), 0.2 mM dNTP mixture (Takara), 1.5 U of AmpliTaq DNA polymerase (Applied Biosystems), 1× PCR buffer II and 2 mM MgCl$_2$ (Applied Biosystems), 0.05 mg L$^{-1}$ bovine serum albumin (Takara), and 3 μL of extracted DNA as the template. PCR was carried out with a GeneAmp9700 (Applied Biosystems) as follows: initial denaturation at 94°C for 5 min; followed by 20 cycles of denaturation at 94°C for 1 min, annealing at 65–55°C (-0.5°C/cycle) for 1 min and extension at 72°C for 1 min; 15 cycles of denaturation at 94°C for 1 min, annealing at 55°C for 1 min and extension at 72°C for 1 min; and a final extension step at 72°C for 7 min. The PCR products were confirmed on a 1.5% agarose gel stained with ethidium bromide.

DGGE analysis was performed as described by Takasu et al. (2011). For each sample, 20 μL of PCR product was applied to each well of an 8% polyacrylamide gel with a denaturing gradient of 25% to 65%. Electrophoresis was performed at a constant voltage of 80 V at 60°C for 15 h in 0.5× TAE buffer. After electrophoresis, gels were stained with SYBR Gold (Molecular Probes) to visualize the bands. Shannon's diversity index ($H'$) for each sample was calculated using the number of the visualized bands and their relative intensity on the stained gel with the following Eq. (1):

$$H' = -\sum(p_i \times \ln p_i) \tag{1}$$

where, $p_i$ is the relative intensity of band $i$ in the sample.




### 2.6 Identification of major bacteria

Major bands on DGGE gels were excised on a UV transilluminator and prepared for DNA sequencing. To extract DNA, excised bands were soaked in 100 μL of TE buffer in 1.5 mL tubes with slow rotation overnight at room temperature. Extracted DNA was re-amplified by PCR with the same protocol given above, except that the reaction mixture volume was 50 μL with 1 μL of DNA template. The PCR products were run on DGGE to confirm that a single DGGE band was amplified, and the PCR products were purified using the QIAquick PCR purification kit (QIAGEN), according to manufacturer's instructions. Purified PCR products were sequenced on an ABI Genetic Analyzer 3130 (Applied Biosystems) with BigDye Terminator. Primer 907r was used for sequencing.

Obtained sequences were aligned and compared with known bacterial sequences in the DNA database of the National Center for Biotechnology Information (NCBI) using the Basic Local Alignment Search Tool (BLAST). Phylogenetic relationships were inferred using the neighbor-joining method (Saitou and Nei, 1987) with 500 bootstrap replications by MEGA6 (Tamura et al., 2013). *Archeoglobus veneficus* SNP6 (gi507148078) was included as the outgroup.

## 3 Results

### 3.1 Prokaryotic cell counts

At the beginning of the both experiments (Day 0), prokaryotic cell counts in FSW were very low, as expected because these were just after filtration through a 0.2 μm filter to prepare the FSW. In all FSW bottles, the counts increased markedly during the course of the incubation, although the timing and rate of increase differed in the different temperatures (Fig. 1): at 25°C (Exp I), the cell counts in the FSW bottle increased rapidly to a level comparable to the level in natural seawater, while cell counts in the lower temperature incubations (6°C) increased after Day 3 in both Exp I and Exp II.

Supposing that there were only small number of prokaryotes and no grazers in the FSW bottles at the beginning of the experiments, the specific growth rates ($\mu$) of prokaryotes at each temperature could be estimated from the change in the number of cells in each FSW bottle. Further, the difference between the changes in cell number in the FSW and UNF bottles indicates the specific grazing mortality rate ($g$). The estimated $\mu$ and $g$ of prokaryotes at each temperature and experiment are shown in Table 1. At 25°C (Exp I), the estimated $\mu$ during the first day was 3.68 d$^{-1}$, giving a doubling time of 4.5 h, while $g$ was estimated to be 3.51 d$^{-1}$ (Table 1). In Exp II, when the seawater temperature 12°C, the estimated $\mu$ and $g$ were 1.20 d$^{-1}$ (doubling time, 13.9 h) and 1.16 d$^{-1}$, respectively. At the colder incubation temperature (6°C), $\mu$ and $g$ during the first 3 days in the both experiments were low, followed by higher growth rates.

### 3.2 Extracellular proteolytic enzyme activity

As a quantification of metabolic activity, extracellular proteolytic enzymes were assayed. In both Exps I and II, samples incubated at original seawater temperature (Figs. 2a and 3a) showed slight increases at Day 1 in the leucine- and alanine-



aminopeptidase activities in the UNF bottles followed by a gradual decrease, while trypsin- and chymotrypsin-type activities showed decreases over the entire period, except for a small second peak on the Day 2-3 of Exp I. In the UNF bottles at lower temperature (Figs 2b and 3b), aminopeptidase activities were increased, while trypsin- and chymotrypsin-type activities were decreased.

In the FSW bottles, increasing of the aminopeptidase activities were coincident with increasing of the number of prokaryotes for both experiments and temperature (Figs. 1, 2c, 2d, 3c, 3d). Trypsin-type activities decreased in all FSW bottles, with a slower decrease at the lower temperature than at the ambient temperature.

In autoclaved seawater as a control, all protease activities were below detection limit during the entire experimental period in both Exps I and II (data not shown).

**3.3 Prokaryotic community structure**

In Exp II, PCR targeting the 16S rRNA gene was conducted for all samples. PCR products were detected for all samples from UNF water bottles, but products were not detected from the FSW bottles on Days 0 and 1 (data not shown). On Day 3 from FSW incubated at 12°C, the PCR products were detected at a level similar to that for the UNF water bottle, but only faint amplification was detected from the FSW bottle incubated at 6°C. On Days 8 and 19, PCR products were detected from

the FSW bottles at levels similar to those in the UNF water bottles. Detection levels correspond to the prokaryotic cell counts, suggesting that the absence of PCR products for the FSW bottle samples at the early stage are due to the lower amounts of bacterial DNA template in these samples.

All samples from which PCR products were obtained were loaded onto DGGE gels, and many bands could be identified, even for FSW bottles (Figs. 4 and 5). For FSW bottles, Shannon's diversity index ($H'$) was lower prior to the increase in

prokaryotic cell number, while diversity was almost the same for UNF water bottles after cell counts increased. Taxonomic groups most closely related to the excised bands were classified in Alphaproteobacteria, Gammaproteobacteria and Flavobacteria, which are known as typical marine bacteria (Table 2). Figures 6 and 7 show phylogenetic dendrograms based on aligned partial 16S rRNA gene sequences of the excised DGGE bands identified in Figs. 4 and 5, respectively, along with sequences from related bacteria that were retrieved from a database. Sequences obtained from the FSW bottles were not

particularly segregated from those isolated from UNF water bottles in the dendrograms.

**4 Discussion**

In general, prokaryotic abundances in aquatic ecosystems are determined by the balance of growth and mortality rates, which are controlled by nutritional availability, protistan grazing and viral lysis. In this study, FSW bottles were prepared with seawater filtered through 0.2 μm filters, meaning that no protists and few prokaryotes were included in the FSW bottles at

the beginning of the experiments. The prokaryotes in the 0.2 μm filtrates are small (starvation) forms of typical marine bacteria, ultramicrobacteria, or prokaryotes with larger but flexible cells that can pass through the pores of the filter.



Although we cannot conclude whether the prokaryotes in 0.2 μm filtrates at the beginning of the experiments were actually small cells or lager flexible cells, these prokaryotes showed rapid growth and appeared as cells that were trapped on the 0.2 μm filter on Day 1 (Exp I, 25°C) or Day 3 (Exp II, 12°C). The rapid increase of the >0.2 μm prokaryotes in FSW bottles could be due to the absence of protistan grazing and reduced competition among heterotrophic prokaryotes due to the

artificially reduced population (due to filtration) in seawater. Li and Dickie (1985) also reported rapid accumulation of [3]H-amino acids and high growth rates of filtered bacteria in water samples passed through 0.2 μm Nuclepore membranes. The estimated prokaryotic growth rate in FSW (Table 1) may represent the potential growth rate of prokaryotes, a value that is usually obscured by protistan grazing in natural aquatic ecosystems.

Haller et al. (1999) investigated the community structure of >0.2 μm and 0.1–0.2 μm bacteria (0.2 μm filterable bacteria) in

the Western Mediterranean Sea by DGGE. They found that the 0.2 μm filterable bacteria were mainly known, typical marine bacteria, and they concluded that the 0.2 μm filterable bacteria in their study were the starvation forms of typical marine bacteria rather than obligate ultramicrobacteria. Vybrial et al. (1999) reported that bacteria cultivated from 0.2 μm filtrate of Mediterranean Sea water were rod shape with a width of 0.4–0.7 μm, and they concluded that the filtered bacteria were starvation forms at the time of filtration. Hood and MacDonell (1987) isolated 0.2 μm filtered bacteria from a subtropical

estuary. They confirmed that cells exposed to low nutrient conditions became very small and that some grew on both oligotrophic and eutrophic media. In our study, the identified bacteria were also typical marine bacteria (Table 2, Figs. 6 and 7). Although the community components of filtered bacteria were selected via filtration from the natural community, the diversity of the community in FSW was not very low (Figs 4 and 5). These bacteria that passed through the 0.2 μm filter at the beginning of the experiment grew in the FSW bottles during the experimental period under conditions of no grazing and

low competition and trapped on 0.2 μm filters. In this study, we did not capture obligate ultramicrobacteria (Hahn et al., 2003), which do not increase in size under any nutritional conditions because we did not use filters with a pore size smaller than 0.2 μm. Our results extend the finding that general marine bacteria can form small or flexible cells in aquatic environments to coastal surface seawater in summer and winter and imply that starvation forms of typical marine bacteria should be ubiquitous not only in very oligotrophic open oceans, but also in temperate coastal surface environments,

irrespective of season. It was suggested that these bacteria should have high growth rates and contribute to the biogeochemical material cycles, although these are typically difficult to evaluate in natural aquatic ecosystems due to tight coupling with grazing effects. The starvation forms of bacteria seem to always be ready for growth in changeable aquatic ecosystems.

In the field of aquatic microbial ecology, dilution experiments (Landry and Hassett, 1982) have been used to estimate the

growth rate of phytoplankton or bacterioplankton, and microzooplankton grazing rate to them. For dilution experiments, unfiltered seawater and filtered seawater prepared by filtration using 0.2 μm filters are mixed at several levels. From our experiment, it is clarified that at 3 days after filtration, 0.2 μm filtered seawater was no longer "particle free", even if kept in a refrigerator (6°C). At the in situ temperature (25°C for Exp I and 12°C for Exp II), bacteria in the filtered seawater




increased rapidly. This observation should be taken into consideration when conducting dilution experiments with incubation durations of more than several hours at in situ temperatures, especially for studies that include estimates of bacteria growth and mortality rates. In twice-filtered seawater (wFSW), increasing microbial activity was not detected (Fig. 3e), which is different from the observation for FSW. For dilution experiments, filtered seawater should be better to prepare via more than two times of filtration.

The time-course of extracellular aminopeptidase activity, especially the activities measured by hydrolysis of Leu-MCA and Ala-MCA, were coincident with changes in prokaryotic cell counts in each bottle. These activities were mostly detected from particulate fraction (fractionated data is not shown), indicating that most extracellular aminopeptidases in the incubation bottles were prokaryotic cell-associated enzymes (ectoenzymes). High aminopeptidase activity seems to reflect the high activity of prokaryotic communities in the bottles.

In contrast, extracellular trypsin-type and chymotrypsin-type activity did not increase with increasing prokaryotic cell number in each experimental bottle. These activities were highest at Day 0, namely in freshly collected natural seawater, and decreased during the course of the incubation in all bottles (Figs. 2 and 3). Trypsin-type activity is typically detected in natural coastal seawater (Obayashi and Suzuki, 2005), especially in the dissolved fraction (Obayashi and Suzuki, 2008). This suggests that dissolved trypsin- and chymotrypsin-type enzymes should be present in each experimental bottle at the start of the present study but were degraded during the experimental period. Although these dissolved enzymes might be produced by microbes in the bottles during incubation, degradation surpassed production. At higher temperature, trypsin-type activity decreased more rapidly, with the decrease of activity being faster in UNF water bottles than in FSW bottles. These results imply that dissolved trypsin-type enzymes in the incubation bottles might be degraded biologically and chemically because the enzymes are proteins. On the other hand, extracellular trypsin- and chymotrypsin-type activity was usually detected with high activity in natural coastal seawater (Obayashi and Suzuki, 2005; 2008). Producers of extracellular trypsin- and chymotrypsin-type enzymes other than prokaryotes might be present in natural aquatic ecosystems but were included at low levels in the small volume of the incubation bottles. Thao et al. (2015) suggested that, besides prokaryotes, ciliates could also be a source of extracellular proteases in seawater. The results of this study might also be an indirect indication of the partial contribution of organisms other than heterotrophic bacteria to the pool of dissolved extracellular proteases in natural seawater.

## 5 Summary

The results of the FSW microcosm experiments suggest the habitual presence of 0.2 μm filterable forms of typical marine bacteria in coastal aquatic ecosystems. Although it was not clarified in this study whether the 0.2 μm filterable bacteria in the raw seawater were actually smaller than 0.2 μm or were flexible enough to pass through the pores of the filter, the number of prokaryotes trapped on the 0.2 μm filter markedly increased in the FSW bottles, corresponding with an increase in aminopeptidase activities. The FSW microcosm incubations with no grazing and low competition revealed high growth





potential and activity of 0.2 μm filterable bacteria, which should be considered to be significant organic matter processing members in aquatic ecosystems. However, decreasing activities of extracellular trypsin- and chymotrypsin-type enzymes in the FSW bottles implies that there are other sources of these enzymes in natural aquatic ecosystems besides heterotrophic prokaryotes.

5   **Acknowledgements**

This work was partially supported by JSPS KAKENHI Grant Numbers JP26450245 and JP24710005.

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





Table 1. Growth rate and grazing mortality of prokaryotic community in Experiments I and II estimated by change in the number of cells ($P_t/P_{t-1}$). Specific growth rate was calculated from the $P_t/P_{t-1}$ in the FSW bottle. Grazing mortality was calculated from the difference of apparent growth rate between the FSW and UNF bottles.

**Experiment I**

| at 25 deg.C | FSW25C | Specific growth rate ($\mu$) | UNF25C | Apparent specific growth rate ($\mu$-$g$) | Specific grazing mortality ($g$) |
|---|---|---|---|---|---|
| | $P_t/P_{t-1}$ | ln($P_t/P_{t-1}$)/day | $P_t/P_{t-1}$ | ln($P_t/P_{t-1}$)/day | |
| Day 0 - Day 1 | 39.84 | 3.68 | 1.19 | 0.17 | 3.51 |
| Day 1 - Day 3 | 1.31 | 0.13 | 0.52 | -0.33 | 0.46 |
| Day 3 - Day 10 | 0.26 | -0.19 | 0.80 | -0.03 | -0.16 |
| at 6 deg.C | FSW6C | Specific growth rate ($\mu$) | UNF6C | Apparent specific growth rate ($\mu$-$g$) | Specific grazing mortality ($g$) |
| | $P_t/P_{t-1}$ | ln($P_t/P_{t-1}$)/day | $P_t/P_{t-1}$ | ln($P_t/P_{t-1}$)/day | |
| Day 0 - Day 1 | 1.00 | 0.00 | 0.91 | -0.10 | 0.10 |
| Day 1 - Day 3 | 1.70 | 0.27 | 1.12 | 0.06 | 0.21 |
| Day 3 - Day 10 | 30.14 | 0.49 | 1.50 | 0.06 | 0.43 |

**Experiment II**

| at 12 deg.C | FSW12C | Specific growth rate ($\mu$) | UNF12C | Apparent specific growth rate ($\mu$-$g$) | Specific grazing mortality ($g$) |
|---|---|---|---|---|---|
| | $P_t/P_{t-1}$ | ln($P_t/P_{t-1}$)/day | $P_t/P_{t-1}$ | ln($P_t/P_{t-1}$)/day | |
| Day 0 - Day 3 | 36.44 | 1.20 | 1.11 | 0.04 | 1.16 |
| Day 3 - Day 8 | 1.66 | 0.10 | 0.67 | -0.08 | 0.18 |
| Day 8 - Day 19 | 1.10 | 0.01 | 0.97 | 0.00 | 0.01 |
| at 6 deg.C | FSW6C | Specific growth rate ($\mu$) | UNF6C | Apparent specific growth rate ($\mu$-$g$) | Specific grazing mortality ($g$) |
| | $P_t/P_{t-1}$ | ln($P_t/P_{t-1}$)/day | $P_t/P_{t-1}$ | ln($P_t/P_{t-1}$)/day | |
| Day 0 - Day 3 | 2.12 | 0.25 | 0.99 | 0.00 | 0.25 |
| Day 3 - Day 8 | 16.58 | 0.56 | 0.77 | -0.05 | 0.61 |
| Day 8 - Day 19 | 1.46 | 0.03 | 0.72 | -0.03 | 0.06 |



Table 2. Sequence similarity of 16S rRNA gene DNA from excised DGGE bands shown in Figs. 4 and 5 and the most closely related known organism.

| Band | Most closely related organism | Sequence similarity | Taxonomic group |
|------|-------------------------------|---------------------|-----------------|
| **Experiment I** | | | |
| G02 | *Lentibacter algarum* | 482/482 (100%) | Proteobacteria; Alphaproteobacteria; Rhodobacterales; Rhodobacteraceae; Lentibacter. |
| G03 | *Roseovarius marinus* | 475/482 (99%) | Proteobacteria; Alphaproteobacteria; Rhodobacterales; Rhodobacteraceae; Roseovarius. |
| G05 | *Roseovarius azorensis* | 471/482 (98%) | Proteobacteria; Alphaproteobacteria; Rhodobacterales; Rhodobacteraceae; Roseovarius. |
| G06 | *Phaeobacter caeruleus* | 470/482 (98%) | Proteobacteria; Alphaproteobacteria; Rhodobacterales; Rhodobacteraceae; Leisingera. |
| G07 | *Neptuniibacter caesariensis* | 504/507 (99%) | Proteobacteria; Gammaproteobacteria; Oceanospirillales; Neptuniibacter. |
| G13 | *Polaribacter reichenbachii* | 513/523 (98%) | Bacteroidetes; Flavobacteriia; Flavobacteriales; Flavobacteriaceae; Polaribacter. |
| G14 | *Polaribacter reichenbachii* | 512/523 (98%) | Bacteroidetes; Flavobacteriia; Flavobacteriales; Flavobacteriaceae; Polaribacter. |
| G18 | *Sulfitobacter pontiacus* | 499/500 (99%) | Proteobacteria; Alphaproteobacteria; Rhodobacterales; Rhodobacteraceae; Sulfitobacter. |
| G19 | *Pseudoalteromonas marina* | 528/528 (100%) | Proteobacteria; Gammaproteobacteria; Alteromonadales; Pseudoalteromonadaceae; Pseudoalteromonas. |
| G22 | *Roseovarius marinus* | 489/503 (97%) | Proteobacteria; Alphaproteobacteria; Rhodobacterales; Rhodobacteraceae; Roseovarius. |
| G23 | *Methylophaga lonarensis* | 510/534 (96%) | Proteobacteria; Gammaproteobacteria; Thiotrichales; Piscirickettsiaceae; Methylophaga. |
| G24 | *Roseovarius azorensis* | 491/502 (98%) | Proteobacteria; Alphaproteobacteria; Rhodobacterales; Rhodobacteraceae; Roseovarius. |
| G25 | *Phaeobacter caeruleus* | 490/502 (98%) | Proteobacteria; Alphaproteobacteria; Rhodobacterales; Rhodobacteraceae; Leisingera. |
| G26 | *Sulfitobacter pontiacus* | 500/502 (99%) | Proteobacteria; Alphaproteobacteria; Rhodobacterales; Rhodobacteraceae; Sulfitobacter. |
| G30 | *Loktanella soesokkakensis* | 491/502 (98%) | Proteobacteria; Alphaproteobacteria; Rhodobacterales; Rhodobacteraceae; Loktanella. |
| G31 | *Donghicola xiamenensis* | 495/503 (98%) | Proteobacteria; Alphaproteobacteria; Rhodobacterales; Rhodobacteraceae; Donghicola. |
| G34 | *Phaeobacter caeruleus* | 493/500 (99%) | Proteobacteria; Alphaproteobacteria; Rhodobacterales; Rhodobacteraceae; Leisingera. |
| G35 | *Thalassococcus halodurans* | 503/503 (100%) | Proteobacteria; Alphaproteobacteria; Rhodobacterales; Rhodobacteraceae; Thalassococcus. |
| **Experiment II** | | | |
| G37 | *Aestuariibacter halophilus* | 495/509 (97%) | Proteobacteria; Gammaproteobacteria; Alteromonadales; Alteromonadaceae; Aestuariibacter. |
| G38 | *Lentibacter algarum* | 482/483 (99%) | Proteobacteria; Alphaproteobacteria; Rhodobacterales; Rhodobacteraceae; Lentibacter. |
| G39 | *Roseovarius marinus* | 476/483 (99%) | Proteobacteria; Alphaproteobacteria; Rhodobacterales; Rhodobacteraceae; Roseovarius. |
| G43 | *Roseovarius crassostreae* | 471/483 (98%) | Proteobacteria; Alphaproteobacteria; Rhodobacterales; Rhodobacteraceae; Roseovarius. |
| G49 | *Ruegeria conchae* | 462/483 (96%) | Proteobacteria; Alphaproteobacteria; Rhodobacterales; Rhodobacteraceae; Ruegeria. |
| G50 | *Roseovarius pacificus* | 470/483 (97%) | Proteobacteria; Alphaproteobacteria; Rhodobacterales; Rhodobacteraceae; Roseovarius. |
| G51 | *Phaeobacter caeruleus* | 472/483 (98%) | Proteobacteria; Alphaproteobacteria; Rhodobacterales; Rhodobacteraceae; Leisingera. |
| G57 | *Aestuariibacter halophilus* | 491/509 (96%) | Proteobacteria; Gammaproteobacteria; Alteromonadales; Alteromonadaceae; Aestuariibacter. |
| G58 | *Lentibacter algarum* | 483/483 (100%) | Proteobacteria; Alphaproteobacteria; Rhodobacterales; Rhodobacteraceae; Lentibacter. |
| G59 | *Roseovarius marinus* | 476/483 (99%) | Proteobacteria; Alphaproteobacteria; Rhodobacterales; Rhodobacteraceae; Roseovarius. |
| G63 | *Roseovarius crassostreae* | 471/483 (98%) | Proteobacteria; Alphaproteobacteria; Rhodobacterales; Rhodobacteraceae; Roseovarius. |
| G65 | *Roseovarius crassostreae* | 471/483 (98%) | Proteobacteria; Alphaproteobacteria; Rhodobacterales; Rhodobacteraceae; Roseovarius. |
| G73 | *Aestuariibacter halophilus* | 495/509 (97%) | Proteobacteria; Gammaproteobacteria; Alteromonadales; Alteromonadaceae; Aestuariibacter. |
| G75 | *Sulfitobacter marinus* | 472/483 (98%) | Proteobacteria; Alphaproteobacteria; Rhodobacterales; Rhodobacteraceae; Sulfitobacter. |
| G76 | *Loktanella rosea* | 477/483 (99%) | Proteobacteria; Alphaproteobacteria; Rhodobacterales; Rhodobacteraceae; Loktanella. |
| G78 | *Colwellia chukchiensis* | 497/509 (98%) | Proteobacteria; Gammaproteobacteria; Alteromonadales; Colwelliaceae; Colwellia. |
| G85 | *Spongiispira norvegica* | 489/509 (96%) | Proteobacteria; Gammaproteobacteria; Oceanospirillales; Spongiispira. |
| G86 | *Aestuariibacter halophilus* | 496/509 (97%) | Proteobacteria; Gammaproteobacteria; Alteromonadales; Alteromonadaceae; Aestuariibacter. |
| G87 | *Aestuariibacter halophilus* | 491/511 (96%) | Proteobacteria; Gammaproteobacteria; Alteromonadales; Alteromonadaceae; Aestuariibacter. |
| G88 | *Roseovarius marinus* | 476/483 (99%) | Proteobacteria; Alphaproteobacteria; Rhodobacterales; Rhodobacteraceae; Roseovarius. |
| G90 | *Lentibacter algarum* | 477/483 (99%) | Proteobacteria; Alphaproteobacteria; Rhodobacterales; Rhodobacteraceae; Lentibacter. |
| G93 | *Loktanella rosea* | 476/483 (99%) | Proteobacteria; Alphaproteobacteria; Rhodobacterales; Rhodobacteraceae; Loktanella. |





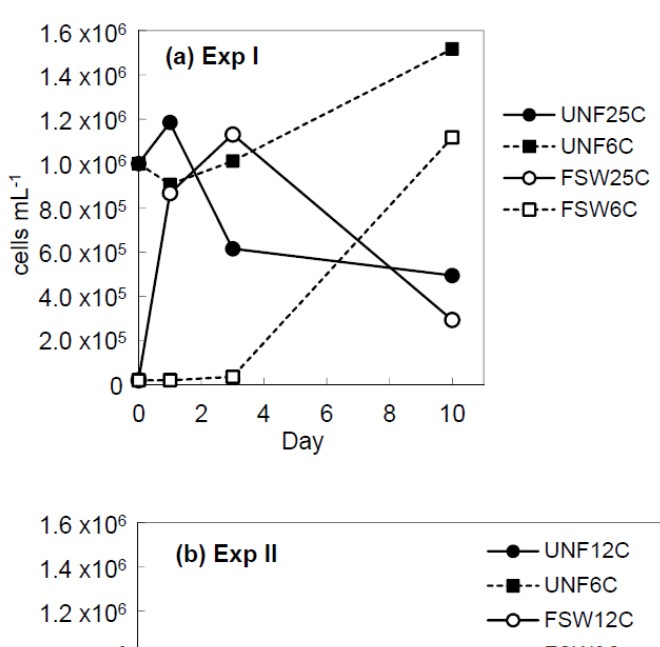

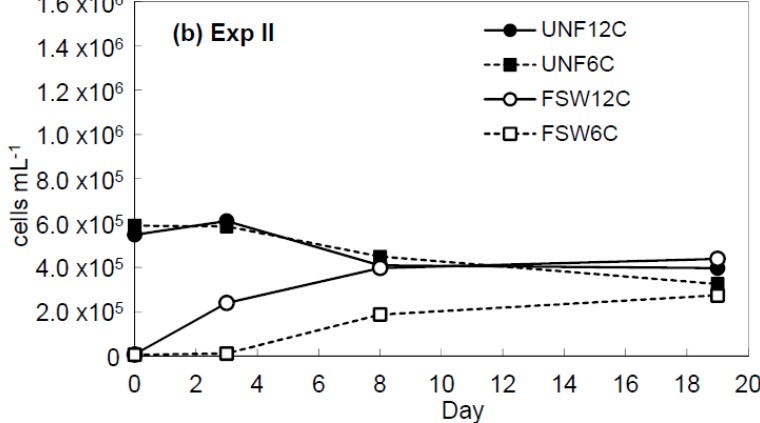

**Figure 1: Number of DAPI-stained prokaryotes trapped on a 0.2 μm filter in (a) Exp I and (b) Exp II.**



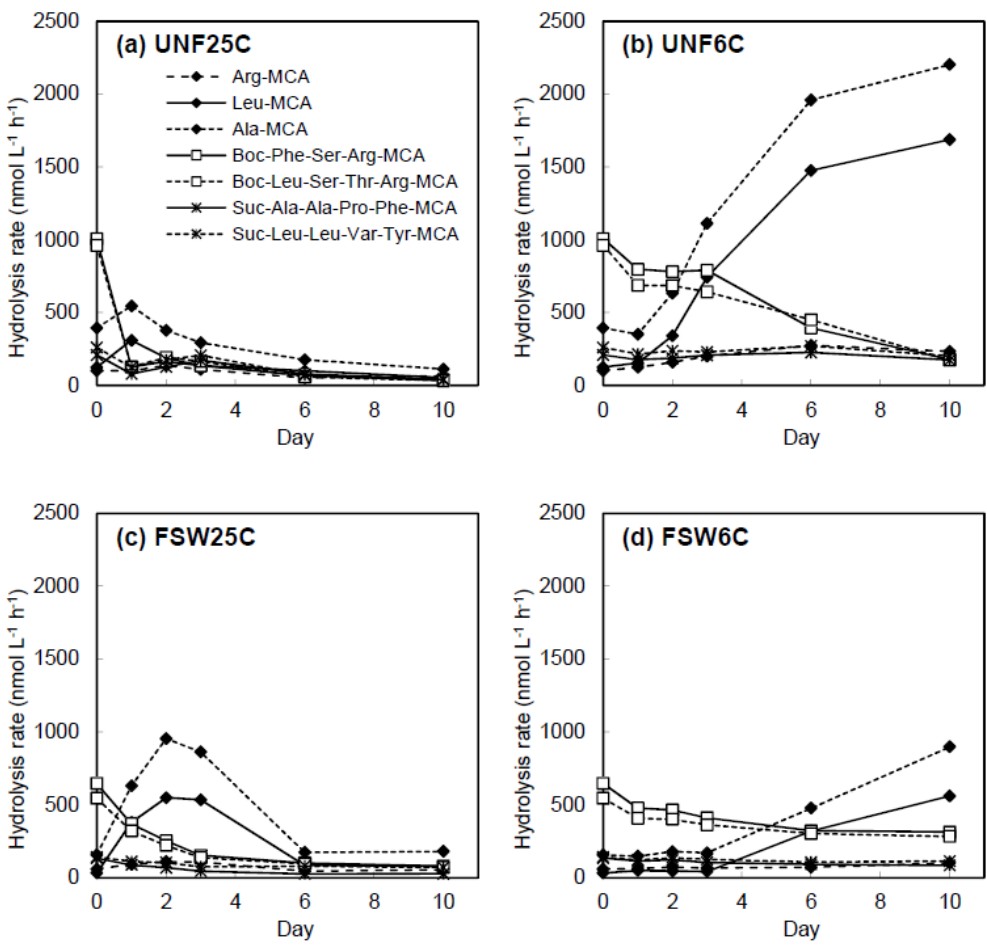

**Figure 2: Extracellular protease activities in microcosm bottles in Exp I. Protease activities were measured using 3 substrates for aminopeptidase (Arg-MCA, Leu-MCA, Ala-MCA), 2 for trypsin-type enzymes (Boc-Phe-Ser-Arg-MCA, Boc-Leu-Ser-Thr-Arg-MCA), and 2 for chymotrypsin-type enzymes (Suc-Ala-Ala-Pro-Phe-MCA, Suc-Leu-Leu-Val-Tyr-MCA).**



**Figure 3: Same as Fig. 2 but for Exp II. The results from the twice-filtered (wFSW) bottles (6°C) are also shown.**





**Figure 4: DGGE fingerprints of bacterial communities in each bottle in Exp I. Excised bands for DNA sequencing and phylogenetic analysis are indicated as small arrowheads with numbers. Shannon's diversity index (H') calculated from the number of bands and relative intensity of each band in each sample is shown in the lower panel.**





**Figure 5: Same as Fig. 4 but for Exp II.**





**Figure 6: Dendrogram of the partial 16S rRNA gene sequences obtained from bands excised from DGGE and known prokaryotic sequences in the database. Labels in the format Gxx correspond to labels in Fig. 4. Labels of sequences from FSW bottles are in bold. The percentages of replicate bootstrap test (500 replicates) are shown next to the branches.**







**Figure 7: Same as Fig. 6 but for Exp II. Gxx labels correspond to labels on bands in Fig. 5.**