# Peer review of "High growth potential and activity of 0.2 $\mu$ m filterable bacteria habitually present in coastal seawater"

_Biogeosciences, 2016_

## Referee Comment (RC1) · Anonymous Referee #1 · 18 Jan 2017

The article of Obayashi and Suzuki describes two long experiments (at different temperatures) focusing on the growth of marine coastal 0.2 $\mu$m filterable bacteria in the absence of grazers (but in the presence of viruses!). As the authors state, the experimental setup was designed to study the lifetime of the dissolved extracellular hydrolytic enzyme activities in seawater.

Due to this change of plan, I do not find the experimental design being optimal to address the high growth potential of 0.2 $\mu$m filterable coastal bacteria. The volume is too small given the long incubation time (19 days). It is not clear how at T0 the authors can estimate the <0.2 $\mu$m bacterial abundance if all or most of the cells escape in the filtrate if they are using 0.2 $\mu$m polycarbonate filters (there is no mention in the text,

that they have used 0.02 $\mu$m alumina oxide filters to solve this issue or alternatively flow cytometry).

Measurements of bacterial volume or bacterial size would have been important to show.

Within the microbial community, fundamental players (that were present in the experimental water, since water has been filtered onto 0.2 $\mu$m) like viruses and their role in shaping the microbial community have not been considered (no viral abundance data, no discussion on their role).

I feel that this manuscript for "non-expert readers" will offer partial and incorrect information on the microbial dynamics that regulate the growth on marine bacteria.

The references are not updated. If the field of ultra-micro bacteria is not popular in these days, grazing and viral lysis are still very hot-topics. Furthermore, given the next-generation sequencing technologies (NGS), it is necessary to compare the DGGE results (and interpretation) with NGS, also if it is challenging.

---

## Referee Comment (RC2) · Anonymous Referee #2 · 13 Feb 2017

General comments: This manuscript examines the growth, protease activity, and taxonomic composition of small bacteria filterable through 0.2 $\mu$m pore size filters. The authors provide excellent background information on the importance of heterotrophic bacterial activities in the ocean, and contextualize the research from the perspective of ultra-small bacteria being examined in the open ocean realms. However, in explaining the novelty of their examinations that focus on coastal systems, they do not provide evidence of why these coastal ecosystems are important in the first place. Further, it is not evident how these small bacteria are potentially influential substrate processers if their presence is only enabled when larger bacteria, which are not classified in the manuscript as competitors, pathogens, or commensals, and grazers are removed. It

is intriguing that the identity of these putatively starved cells are not seemingly different than that of the original community, but the application of DGGE to make this conclusion likely introduces a large level of uncertainty in truly elucidating differences in community composition between and among the manipulations, particularly in light of the current use of next generation sequencing and metagenomic approaches that can identify even rare members of the microbial biosphere, including a new phyla of ultra-small bacteria (Candidate Phyla Radiation).

Specific comments: Page 3, Lines 13-17 – More details on the microscopy counts are required. Were the filters replicated, how many fields of view were counted, minimum number of cells counted per field of view, and approximately how many total cells, etc.?

Page 4, line – 13 – Has it been shown that PCR-DGGE is comparable to other, more recent community composition assessment techniques, e.g. versus iTag, or even versus TRFLP, clone libraries, or others?

Page 5, Line 2 – How many bands were excised? Why were many ignored? There were several bands in figures 4 & 5 that are present in one sample but not in the other. A comprehensive analysis of the community must be performed if 16S iTag sequencing is not performed.

Page 5, Line 15 – Specific numbers are needed to state exactly how low the counts in FSW are.

Page 5, line 20 – If you assume a minimal influence of grazers on growth rate, you must also make an assumption on viruses. Are these assumptions valid? This requires a citation or other explanation.

Page 7, lines 4 – 8 – It is unclear what this section is attempting to support and/or conclude. There should be a citation regarding the influence of the grazer, another section discussing the potential influences of viruses, and a discussion of why the Li & Dickie 1985 paper is mentioned and what it means (particularly since this paper

seems to have already shown what the authors say they are reporting for the first time!). Lastly, it is difficult and likely incorrect to assign a bacterial community growth rate since organic matter, viruses, competition, and a multitude of other abiotic factors affect growth of single cells comprising that community. If the authors believe a potential growth rate is warranted here, a more in depth discussion with citations are needed.

Page 7, lines 17 – 18 – The potential for PCR bias and how it potentially alters the interpretation of community diversity should be mentioned and discussed. Also, is there a statistical method to compare the Shannon indices at each time point?

Page 7, lines 22 – 28 – It is interesting that these cells seem pervasive in coastal waters. However, there is no convincing evidence that they meaningfully contribute to biogeochemical cycles under natural conditions. There must be a discussion outlining what types of conditions the 0.2 $\mu$m filtering mimicked to facilitate increased growth and activity of these cells. This is particularly needed since DAPI counts were not completed everyday, as was the case for the enzyme activity measurements.

Page 8, lines 7 – 8 – If the enzyme activities of the particulate fraction are going to be mentioned, the methods involved with this measurement and the associated data must be included.

Page 8, line 26 – What about the effect of bacterial cell breakage?

Page 9, lines 1 – 2 – As mentioned previously, more evidence is needed to show that 0.2 $\mu$m filterable bacteria are significantly contributing to organic matter biogeochemistry. This is difficult to reconcile with the fact that the DGGE-based conclusions that state the small bacteria are no different than the unfiltered community toward the end of the microcosm experiments. How can the community be the same yet occupy a different biogeochemical niche? The results seem to suggest a succession in community composition that is dictated by organic matter availability and/or competition at the early stages among the small bacteria. In any case, more thought out conclusions must be formulated that include a mechanistic model of how these manipulated conditions reflects actual microscale events that are likely to influence microbial ecological control of biogeochemistry.

Technical corrections Page 2, lines 19 – 20 – This sentence should be rewritten for clarification.

Page 4, line 28 – Change to Arabic numerals.

Page 5, lines 15 – 16 – Rewrite for clarification.

Page 7, lines 29 – 30 – Rewrite for clarification.

Page 8, lines 3 – 5 – Rewrite both sentences for clarification.

---

## Author Comment (AC1) · 7 Mar 2017

Thank you very much for the helpful comments. Here is our reply to each comment.

—Comment— The article of Obayashi and Suzuki describes two long experiments (at different temperatures) focusing on the growth of marine coastal 0.2 $\mu$m filterable bacteria in the absence of grazers (but in the presence of viruses!). As the authors state, the experimental setup was designed to study the lifetime of the dissolved extracellular hydrolytic enzyme activities in seawater. Due to this change of plan, I do not find the experimental design being optimal to address the high growth potential of 0.2 $\mu$m filterable coastal bacteria. The volume is too small given the long incubation time (19 days). It is not clear how at T0 the authors can estimate the <0.2 $\mu$m bacterial abundance if all

or most of the cells escape in the filtrate if they are using 0.2 $\mu$m polycarbonate filters (there is no mention in the text, that they have used 0.02 $\mu$m alumina oxide filters to solve this issue or alternatively flow cytometry). Measurements of bacterial volume or bacterial size would have been important to show.

—Reply—

As the Referee pointed out, our experiments might not be optimized way to show the growth of 0.2 $\mu$m filterable bacteria directly, however, the results of our experiments which demonstrated rapid increase of the number of prokaryotes trapped on 0.2 $\mu$m filter in the FSW bottle (originally 0.2 $\mu$m filtered seawater) also show the evidence of high growth potential of filterable bacteria in seawater.

We used 1-L polycarbonate bottles for the experiments because of the size limitation of the incubators (to incubate many bottles in the same condition). The actual starting volume of the water in each bottle was about 1.2 L, and more than half of the water still remained at the end of the experiments.

In this study, we did not estimate abundances of the "actual <0.2 $\mu$m" cells and viruses in the sample during the experiments, so that we were not able to mention "obligate ultramicrobacteria". In the experiments, the number of prokaryotes was counted on 0.2 $\mu$m filter after filtration (usually 2 mL) and the results demonstrated rapid increase of the number of the >0.2 $\mu$m cells in the FSW bottles. For Day 0 sample of FSW, 3 mL of the water sample was filtered onto black polycarbonate 0.2 $\mu$m filter after staining with DAPI. More than 200 fields were observed under fluorescent microscope (x1000) and very few cells were found on the filter. Although the estimated cell numbers on Day 0 in FSW could include much uncertainties, it was clear that the number of cells on Day 0 in FSW was much lower (almost 2 orders of magnitude lower) than those on later days of FSW bottles and those in UNF bottles.

All bacteria we counted here were larger than 0.2 $\mu$m.
—Comment— Within the microbial community, fundamental players (that were present in the experimental water, since water has been filtered onto 0.2 $\mu$m) like viruses and their role in shaping the microbial community have not been considered (no viral abundance data, no discussion on their role). I feel that this manuscript for "non-expert readers" will offer partial and incorrect information on the microbial dynamics that regulate the growth on marine bacteria.

—Reply—

As the Referee mentioned, viruses are also fundamental players in the microbial community. In our experiment, virus should be abundant both in FSW bottles and UNF bottles. We supposed that the viruses were similarly abundant in the both bottles at the beginning of the experiments and that the differences of the microbial community between in FSW and in UNF at the beginning of the experiments were the abundances of grazers and prokaryotes; no grazers and much lower abundance of prokaryotes in FSW bottles compared to UNF bottles. It can be supposed that the increasing of the number of prokaryotes in FSW bottles during the early stage of the experiments were attributed to no grazers and low competition condition for prokaryotes. The "seeds" of the increasing prokaryotes in FSW bottles should be filterable bacteria present in the FSW (0.2 $\mu$m filtrate) at the beginning of the experiments. On the other hand, decreasing of the number of prokaryotes during the later stage of the experiments should be due to the effect of viruses. We would like to add mentions of viruses and their roles in discussion part of the revised version of our manuscript.

—Comment— The references are not updated. If the field of ultra-micro bacteria is not popular in these days, grazing and viral lysis are still very hot-topics.

—Reply—

Thank you for the helpful comment about the references. We will update references especially for those about microbial dynamics in aquatic environments in the revised manuscript.

—Comment— Furthermore, given the next-generation sequencing technologies (NGS), it is necessary to compare the DGGE results (and interpretation) with NGS, also if it is challenging.

—Reply—

As the Referee said, NGS analysis is powerful tool to see all members in sample, whereas DGGE can selectively show abundant members. This is a rather merit of DGGE. The information from DGGE profiles combination with sequencing of the bands could provide visualized information about the abundant bacterial community (>0.2 $\mu$m) reconstructed in FSW bottles from the "seeds" in 0.2 $\mu$m filtrates. The results of DGGE do not represent whole structure of bacterial community, however, each bacterium which was detected in the DGGE was rightly present in the sample. From the results that typical marine bacteria (Alphaproteobacteria, Gammaproteobacteria, and Flavobacteria) were detected from FSW microcosms, we can say that the "seeds" of these bacteria were existed in 0.2 $\mu$m filtrates at the beginning of the experiments, indicating that these bacteria with filterable form (small or flexible enough to pass through 0.2 $\mu$m filter) were present in the original coastal seawater.

---

## Author Comment (AC2) · 8 Mar 2017

Thank you very much for the helpful comments. Here is our reply to each comment. Some parts of replies overlapped among the comments.

—Comment— This manuscript examines the growth, protease activity, and taxonomic composition of small bacteria filterable through 0.2 $\mu$m pore size filters. The authors provide excellent background information on the importance of heterotrophic bacterial activities in the ocean, and contextualize the research from the perspective of ultra-small bacteria being examined in the open ocean realms. However, in explaining the novelty of their examinations that focus on coastal systems, they do not provide evidence of why these coastal ecosystems are important in the first place.

[Figure]

—Reply—

Thank you for the comments. Filterable bacteria in aquatic environment have been reported as a starvation form in oligotrophic environment. Temperate coastal seawater, such as our observation field, is usually not oligotrophic. In this manuscript we intend to show that filterable bacteria are habitually present not only in oligotrophic water but also in non-oligotrophic environment such as temperate coastal ecosystems that we examined. We would like to rewrite a part of introduction to make clear the meaning of the study dealt with filterable bacteria in coastal seawater in the revised manuscript.

—Comment— Further, it is not evident how these small bacteria are potentially influential substrate processers if their presence is only enabled when larger bacteria, which are not classified in the manuscript as competitors, pathogens, or commensals, and grazers are removed.

—Reply—

In this study, we did not see abundances and community structure of the "actual <0.2 $\mu$m" cells in the sample during the experiments, so that we were not able to mention the structure of actual <0.2 $\mu$m bacterial community and "obligate ultramicrobacteria". However, the facts that increase of the number of the >0.2 $\mu$m bacteria in the FSW bottles indicated that at least a part of filterable bacteria should be "seeds" of these >0.2 $\mu$m bacteria. The results that these >0.2 $\mu$m bacteria appearing and detected in FSW bottles were typical marine bacteria (Alphaproteobacteria, Gammaproteobacteria, and Flavobacteria) suggested that filterable forms (small or flexible enough to pass through 0.2 $\mu$m filter) of these typical marine bacteria were present in the original coastal seawater. The rapid increase of these bacteria and corresponding elevation of aminopeptidase activity in FSW meant that the "seeds" have ability to utilize organic matter in seawater for their growth. From the similar results of two experiments (Exp I in summer and Exp II in winter), we conclude that 0.2 $\mu$m filterable bacteria habitually exist in coastal non-oligotrophic seawater and they have potential to metabolize

organic matter biogeochemically.

—Comment— It is intriguing that the identity of these putatively starved cells are not seemingly different than that of the original community, but the application of DGGE to make this conclusion likely introduces a large level of uncertainty in truly elucidating differences in community composition between and among the manipulations, particularly in light of the current use of next generation sequencing and metagenomic approaches that can identify even rare members of the microbial biosphere, including a new phyla of ultra-small bacteria (Candidate Phyla Radiation).

—Reply—

As the Referee said, metagenomic approach using NGS analysis is suitable to know the whole structure of bacterial community in each microcosm and to compare those structures among the microcosms, while DGGE can selectively show abundant members. The DGGE profiles combined with sequences of the bands provided some visualized information about the bacterial (>0.2 $\mu$m) community reconstructed in FSW bottles from the "seeds" in 0.2 $\mu$m filtrates. The results of DGGE do not represent whole structure of bacterial community, however, each bacterium which was detected in the DGGE and sequencing analysis was rightly present in the sample. From the results that typical marine bacteria were detected from FSW microcosms, we can say that the "seeds" of these bacteria existed in 0.2 $\mu$m filtrates at the beginning of the experiments, indicating that these bacteria with filterable form were present in the original coastal seawater.

From the point of view that DGGE profiles do not represent whole structure of bacterial community, we reconsidered the discussion and realized that the comparison of community diversity based on Shannon's diversity indices (H') calculated from DGGE band profiles was not suitable. We would like to remove the H' data and related discussion on diversity of bacterial community from our manuscript in revised version.

—Comment— Page 3, Lines 13-17 – More details on the microscopy counts are required. Were the filters replicated, how many fields of view were counted, minimum number of cells counted per field of view, and approximately how many total cells, etc.?

—Reply—

We will describe more details about prokaryotic cell counting under the microscope in the revised version of the manuscript. Duplicate filters were prepared for each sample. Under microscope, 100~300 fields were counted on a filter. Total cells counted on a filter were usually more than 1500 cells, except for the sample that contained very low number of cells (the early days of FSW bottles). For the early days of FSW, more than 200 fields were observed and very few cells were found on the filter. Although the estimated cell numbers in these samples could include much uncertainties, it was clear that the number of cells in early days of FSW was much lower (almost 2 orders of magnitude lower) than those on later days of FSW bottles and those in UNF bottles.

—Comment— Page 4, line – 13 – Has it been shown that PCR-DGGE is comparable to other, more recent community composition assessment techniques, e.g. versus iTag, or even versus TRFLP, clone libraries, or others?

—Reply—

We did not conduct other assessment for bacterial community composition at this time.

—Comment— Page 5, Line 2 – How many bands were excised? Why were many ignored? There were several bands in figures 4 & 5 that are present in one sample but not in the other. A comprehensive analysis of the community must be performed if 16S iTag sequencing is not performed.

—Reply—

Actually 36 bands from Exp I and 57 bands from Exp II, which were thought to be important, were excised and tried to determine their sequences, however, parts of them were not successful because of unsuitable concentration or purity of DNA in

the excised band or other technical uncertainties. Indications on Figures 4 and 5, and phylogenetic analysis (Figure 6 and 7) included only the bands which obtained reliable sequence data. As I mentioned above, even though the results of DGGE did not represent whole community of bacteria, at least each bacterium which was detected in the DGGE and sequencing analysis was rightly present in the sample.

—Comment— Page 5, Line 15 – Specific numbers are needed to state exactly how low the counts in FSW are.

—Reply—

We will add them in the revised manuscript. The estimated numbers were around 1e4 cells/mL. As I mentioned above, these estimation might include some uncertainties because of the lower number of cells, however, it was clear that the number of cells in early days of FSW was much lower than those on later days of FSW bottles and those in UNF bottles.

—Comment— Page 5, line 20 – If you assume a minimal influence of grazers on growth rate, you must also make an assumption on viruses. Are these assumptions valid? This requires a citation or other explanation.

—Reply—

Thank you for the comment. In our experiment, virus should be abundant both in FSW bottles and UNF bottles. We supposed that the viruses were similarly abundant in the both bottles at the beginning of the experiments and that the differences of the microbial community between in FSW and in UNF at the beginning of the experiments were the abundances of protists and prokaryotes; no protists and much lower abundance of prokaryotes in FSW bottles compared to UNF bottles. It can be assumed that the increasing of the number of prokaryotes in FSW bottles during the early stage of the experiments was attributed to no grazers and low abundance of prokaryotes. However, now we reconsidered that the condition was different in later stage of the experiment.

From this point, we would like to delete Table 1 and rewrite the description of the results on page 5 line 20-, and add the mentions of viruses and their roles in discussion part of the revised version of our manuscript.

—Comment— Page 7, lines 4 – 8 – It is unclear what this section is attempting to support and/or conclude. There should be a citation regarding the influence of the grazer, another section discussing the potential influences of viruses, and a discussion of why the Li & Dickie 1985 paper is mentioned and what it means (particularly since this paper seems to have already shown what the authors say they are reporting for the first time!). Lastly, it is difficult and likely incorrect to assign a bacterial community growth rate since organic matter, viruses, competition, and a multitude of other abiotic factors affect growth of single cells comprising that community. If the authors believe a potential growth rate is warranted here, a more in depth discussion with citations are needed.

—Reply—

As I mentioned above, we reconsidered about the growth rate. We will rewrite this part in revised manuscript.

—Comment— Page 7, lines 17 – 18 – The potential for PCR bias and how it potentially alters the interpretation of community diversity should be mentioned and discussed. Also, is there a statistical method to compare the Shannon indices at each time point?

—Reply—

Thank you for the comment. The potential for PCR bias will be added in discussion in the revised manuscript.

As I mentioned above, we reconsidered that comparison of the community diversities based on the result of PCR-DGGE was not suitable. We would like to remove Shannon indices (lower panels in Figures 4 and 5) and related description in the revised manuscript.

—Comment— Page 7, lines 22 – 28 – It is interesting that these cells seem pervasive in coastal waters. However, there is no convincing evidence that they meaningfully contribute to biogeochemical cycles under natural conditions. There must be a discussion outlining what types of conditions the 0.2 $\mu$m filtering mimicked to facilitate increased growth and activity of these cells. This is particularly needed since DAPI counts were not completed everyday, as was the case for the enzyme activity measurements.

—Reply—

In our experiment, we prepared FSW microcosms (0.2 $\mu$m filtrates) and UNF microcosms (unfiltered seawater). We supposed that the viruses were similarly abundant in the both bottles at the beginning of the experiments and that the differences of the microbial community between in FSW and in UNF at the beginning of the experiments were the abundances of protists and prokaryotes. It can be supposed that the increasing of the number of prokaryotes in FSW bottles during the early stage of the experiments were attributed to no grazing and low competition condition for prokaryotes. Actually, it is hard to think that completely "no grazers" condition occurs in natural environments, however, grazing pressure could be fluctuated in aquatic environments. Considering that the results of DGGE combined with band sequences detected many phylogenetic groups of typical marine bacteria in reconstructed community in FSW bottles, it can be thought that many bacteria have high growth potential and ready for growing when they face to favorable conditions. The elevation of ectoenzymatic aminopeptidase activities in FSW bottles was corresponding to the DAPI counts and was also supportive data indicating the high growth potential of bacteria. We would like to include the data of aminopeptidase activity with fractionation in the revised manuscript, as the reviewer suggestion in the next comment.

—Comment— Page 8, lines 7 – 8 – If the enzyme activities of the particulate fraction are going to be mentioned, the methods involved with this measurement and the associated data must be included.

—Reply—

We will add some data of enzyme activities with fractionation and its method in the revised manuscript.

—Comment— Page 8, line 26 – What about the effect of bacterial cell breakage?

—Reply—

Yes, bacterial cell breakage due to grazing or viral lysis should be parts of natural processes to provide dissolved enzymes in seawater. The phrase in our manuscript "partial contribution of organisms other than heterotrophic bacteria" (page 8 line 25) was not enough to explain. We will rewrite this sentence to include not only direct production by other organisms but also releasing of these enzymes via interaction between bacteria and other organisms (e.g. bacterial cell breakage accompanying with grazing process).

—Comment— Page 9, lines 1 – 2 – As mentioned previously, more evidence is needed to show that 0.2 $\mu$m filterable bacteria are significantly contributing to organic matter biogeochemistry. This is difficult to reconcile with the fact that the DGGE-based conclusions that state the small bacteria are no different than the unfiltered community toward the end of the microcosm experiments. How can the community be the same yet occupy a different biogeochemical niche? The results seem to suggest a succession in community composition that is dictated by organic matter availability and/or competition at the early stages among the small bacteria. In any case, more thought out conclusions must be formulated that include a mechanistic model of how these manipulated conditions reflects actual microscale events that are likely to influence microbial ecological control of biogeochemistry.

—Reply—

As I mentioned above, the results of DGGE might not represent whole structure of bacterial community, however, each bacterium which was detected in the DGGE and

sequencing analysis was rightly present in the sample. Detection of typical marine bacteria (Alphaproteobacteria, Gammaproteobacteria, and Flavobacteria) from the reconstructed community (>0.2 $\mu$m) in FSW microcosms indicated at least that the "seeds" of these bacteria should exist in 0.2 $\mu$m filtrates at the beginning of the experiments. And this result suggests that these bacteria with filterable forms (small or flexible enough to pass through 0.2 $\mu$m filter) were present in the original coastal seawater. Moreover, the rapid increase of these bacteria and corresponding elevation of aminopeptidase activity in FSW meant that the "seeds" have ability to utilize organic matter in seawater for their growth. These results suggest that filterable bacteria are habitually present in coastal seawater, even in non-oligotrophic environment, and they seem to have potential to process organic matter in seawater biogeochemically.

—Comment— Technical corrections Page 2, lines 19 – 20 – This sentence should be rewritten for clarification. Page 4, line 28 – Change to Arabic numerals. Page 5, lines 15 – 16 – Rewrite for clarification. Page 7, lines 29 – 30 – Rewrite for clarification. Page 8, lines 3 – 5 – Rewrite both sentences for clarification.

—Reply—

Thank you very much for these technical comments. Page 4 line 28 will not be included in revised manuscript. We will rewrite other parts to make the meaning clear in our revised manuscript.

―――――――――――――――――